# Lipidome visualisation, comparison, and analysis in a vector space

Timur Olzhabaev[1,2☺], Lukas Müller[1,2☺], Daniel Krause[2], Dominik Schwudke[2,3,4‡], Andrew Ernest Torda[1‡*]

1 Centre for Bioinformatics, University of Hamburg, Hamburg, Germany, 2 Bioanalytical Chemistry, Research Center Borstel Leibniz Lung Center, Borstel, Germany, 3 German Center for Infection Research, Thematic Translational Unit Tuberculosis, Borstel, Germany, 4 German Center for Lung Research (DZL), Airway Research Center North (ARCN), Borstel, Germany

☺ These authors contributed equally to this work.
‡ DS and AET Joint Senior Authors
* andrew.torda@uni-hamburg.de

## Abstract

A shallow neural network was used to embed lipid structures in a 2- or 3-dimensional space with the goal that structurally similar species have similar vectors. Tests on complete lipid databanks show that the method automatically produces distributions which follow conventional lipid classifications. The embedding is accompanied by the web-based software, Lipidome Projector. This displays user lipidomes as 2D or 3D scatterplots for quick exploratory analysis, quantitative comparison and interpretation at a structural level. Examples of published data sets were used for a qualitative comparison with literature interpretation.

## Author summary

Lipids are not just the basis of membranes. They carry signals and metabolic energy. This means that the presence, absence, and quantity of lipids reflects a cell's biochemical state - starving, nourished, sick or healthy. Lipidomics (measuring all lipids in a biological specimen) provides lists of the chemical species and their quantities.

We have used a shallow neural network from natural language modelling to embed lipids in a continuous vector space. Firstly, this means that similar molecules have similar positions in this space. Conventional lipid categories cluster automatically. Secondly, the accompanying web-based software, Lipidome Projector imports a lipidome and displays it as a set of points. Reading several lipidomes at once allows quantitative and structural comparisons. Combined with the ability to show structure and abundance diagrams, the software allows exploratory analysis and interpretation of lipidomics datasets.

## Introduction

Lipids remind one of membranes or fats, but they also carry energy and signals, so one may assume that the set of lipids in a sample reflects the health and metabolic state of a tissue or

**Data availability statement:** All relevant data are within the manuscript, its supporting information files, and the Lipidome Projector GitHub repository (https://www.github.com/olzhabaev/lipidome_projector). Lipidome Projector is released under the MIT license. It is a web-application that can be run locally or deployed to a server. The repository has pre-computed vectors for and pre-parsed versions of the Lipid Maps and SwissLipids databases. The software distribution also includes modules for the pre-processing of the databases and a complete recalculation of the vector space. An instance of Lipidome Projector is available at: https://lipidomeprojector.zbh.uni-hamburg.de/.

**Funding:** We acknowledge support from the German Network For Bioinformatics Infrastructure (de.NBI, https://www.fz-juelich.de/en/ibg/ibg-5/networks/de.nbi, Contract-number: W-de.NBI-006) to DS. TO and LM received salaries from de.NBI. The funders had no role in study design, data collection and analysis, decision to publish, or preparation of the manuscript.

**Competing interests:** The authors have declared that no competing interests exist.

organism. Mass spectrometry provides lipidome information, but a list of $10^2$-$10^4$ lipids and their quantities is not easily interpretable. For exploratory analysis, one would like a method that highlights chemical trends and shows how samples differ with respect to lipid structures and quantities. Given a set of mass spectrometry peaks that have been assigned to lipids, the idea is to display lipidomes as scatterplots in a 2- or 3-dimensional space. This requires two steps. First, there must be a continuous vector space such that each lipid gets distinct coordinates. Second, one needs software to display and compare plots interactively. The software should make it easy to relate points back to their names and chemical structures.

The aims here are different to those of other lipidomics software packages. If one wants to treat a lipidome similarly to gene expression data, one can look for changed levels of lipids or focus on molecules whose abundances are correlated [1–3]. If one wants to see a lipidome in terms of networks, there is network construction and display software [4]. Our focus is different. Lipidome Projector lets one quickly highlight and interactively explore differences between groups of samples, with the simultaneous display of abundances and structures.

The first challenge is finding vectors for molecules for the two- and three-dimensional plots. Previous attempts applied ideas from string comparisons [5], but this was not without problems. Whatever notation one uses, a small change to a molecule can lead to a large change in a string representation such as SMILES [6], so the similarity metrics are fundamentally unstable. Kopczynski et al approached the problem with elegant distance metrics, but this required some preconceptions about lipid structures and used expensive graph similarity methods [7].

We come to the problem with slightly different ideas and some specific goals. The method should be objective, unsupervised and require minimal chemical preconceptions. Coordinates should be quite different for unrelated molecules, but systematic changes such as extending the length of an aliphatic chain should give a series of points near each other. Adding a phosphate or alcohol group to two different molecules should change both coordinates in a similar manner. Our method for lipids is a reimplemented and adjusted version of Mol2Vec [8], a technique from the small-molecule literature which is, in turn, based on Word2Vec [9] a word embedding method from natural language processing. To embed words, one first defines a vocabulary and gives each word a unique token. In a text corpus, similar tokens appear in similar contexts with reasonable probability, such that a token/ context prediction task can be used to train semantic vector representations. To apply the idea in chemistry, one constructs a vocabulary of chemical fragments and trains a shallow network on a large set of molecules to recognise surrounding contexts. Input fragments are represented by integer identifiers derived from computed sparse connectivity fingerprints [10]. Fragment vectors come from hidden layer weights of the trained network and are summed to produce vector representations of entire molecules.

Calculating the vector space model is performed once on a large set of lipid structures and takes several hours. User lipidome data is simply matched to precomputed vectors. Lipidome Projector, the browser-based application for visualization and analysis, allows one to interactively explore lipidomes in the vector space and additionally displays lipid abundance charts and molecular structures.

To judge our methods, we consider the distributions of lipids in the computed vector space and apply Lipidome Projector visualizations on three published lipidome datasets.

## Materials and methods

### Lipid vector space

For training, the Lipid Maps Structure Database (LMSD) [11] and SwissLipids [12] (both accessed Jan 2023) were combined. SwissLipids entries were filtered to obtain lipids with valid

SMILES at isomeric subspecies level. The combination of databases resulted in over 620 000 unique structures. RDKit [13] was used to convert all database entries to a consistent charge state and RDKit's implementation of extended connectivity fingerprints [10] was used to assign a unique identifier to each substructure of a specified radius around each atom. Substructure identifiers were ordered according to the position of the substructure's central atom within the molecule's canonical SMILES string.

Our implementation makes a few necessary modifications to Mol2Vec's model. RDKit's function for the computation of fingerprints for the generation of substructure identifiers was parameterised to use chirality. No rare substructures were filtered or replaced by a special token. Finally, a parameter had to be adapted to capture differences in long alkyl chains. Mol2Vec descriptors for small molecules were built from fragments using atoms (radius 0) and their immediate neighbours (radius 1). For our much larger lipid structures, radii of size 0, 1, 2, 3, 4, 5, 10, 15, 20, 25, 30, 35, 40, 45 and 50 were used, resulting in just under three million unique fragments for the combination of databases. This means, that for each lipid, the set of fragments for each radius had to be used as a separate training sentence to avoid fragments of vastly different sizes being put together as training pairs in Word2Vec's sliding window training data generation.

Gensim [14] was used to train the Word2Vec model with training parameters listed in S1 Table. The network generated 100-dimensional substructure vectors, which were summed for each molecule. For visualization, the Barnes-Hut [15] version of t-distributed stochastic neighbour embedding [16] as implemented in OpenTSNE [17] was used to reduce the 100-dimensional vector space to 2- and 3-dimensional vector sets. PCA initialization was used to improve reproducibility and attempt to preserve global structure [18] (the remaining significant parameters are in S2 Table). The embedding process is summarised in Fig 1A.

## Lipidome processing

As part of building the system, entries from the lipid databases are stored along with their corresponding vectors and higher-level abbreviations for each isomer following previously defined levels [19]. When a user lipidome is imported, entries are matched against pre-calculated vectors (Fig 1B). Goslin [20] is used to parse both databases and user data. It accepts common nomenclature, but should it fail, the process will look for a match based on user-provided names. This means that Lipidome Projector covers at least all entries from the union of SwissLipids and the LMSD that were successfully parsed by Goslin (S1 Dataset gives a list of translated class names).

Mass spectrometry often does not identify a lipid at the complete structure level [19] so additional steps are necessary to deal with this ambiguity. The software finds the set of isomers that match the higher-level abbreviation, but not all members of this set will be plausible for the organism under consideration. To filter the list of possible lipids, Lipidome Projector expects a constraints list with allowed fatty acyls and long-chain bases. The remaining isomer vectors are averaged to produce a single representative vector.

## Visualization and analysis software

Plots are generated using Plotly.py [21]. Marker sizes are derived from lipid abundances, to which either linear or min-max scaling is applied. Dash [21] was used to build the web-application front end. The rest of the application was built in Python [22] with pandas [23] used for data-table storage and manipulation. Parsing and matching are performed server-side. The original lipidome dataset together with the newly derived lipid names and computed vectors is stored inside the user's browser session and sent to the server for temporary processing operations such as averaging of sample groups, computation of Log2FC values between samples, or plot updates. Lipidome datasets and constraints are read in a simple table format.

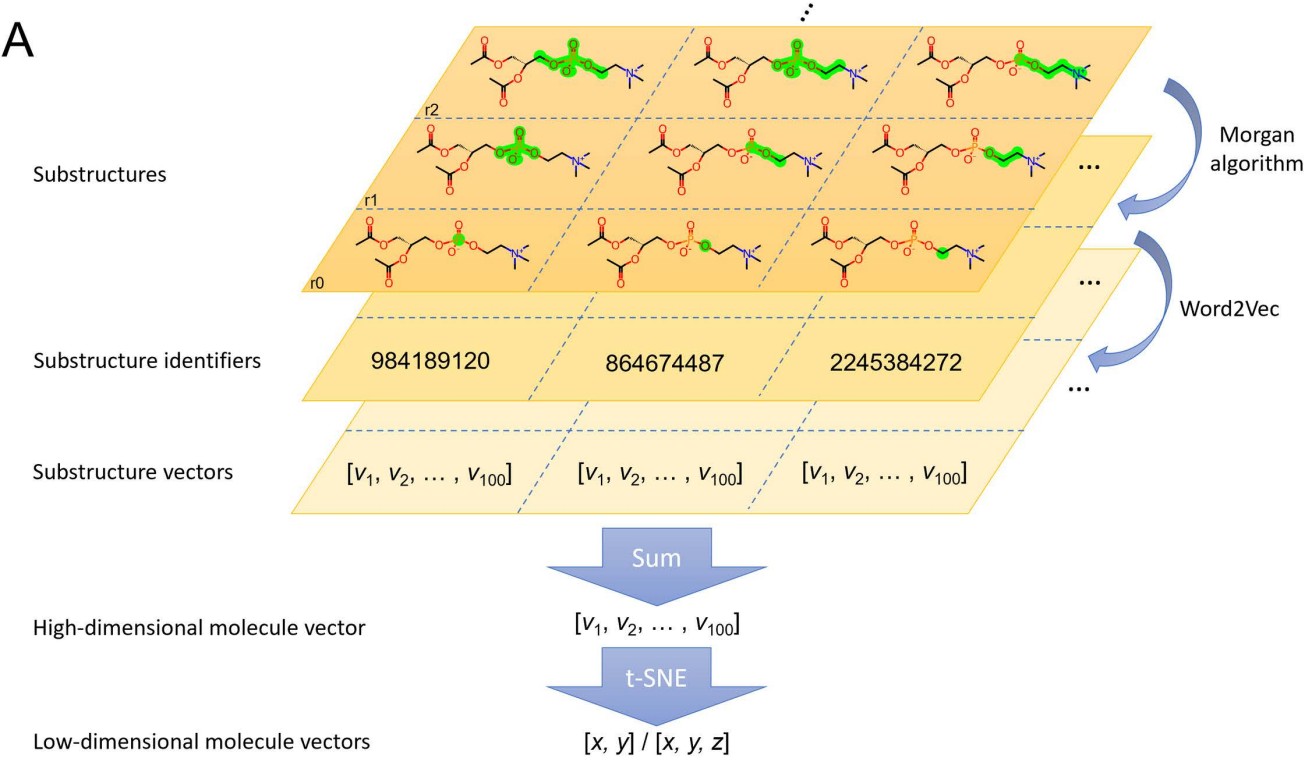

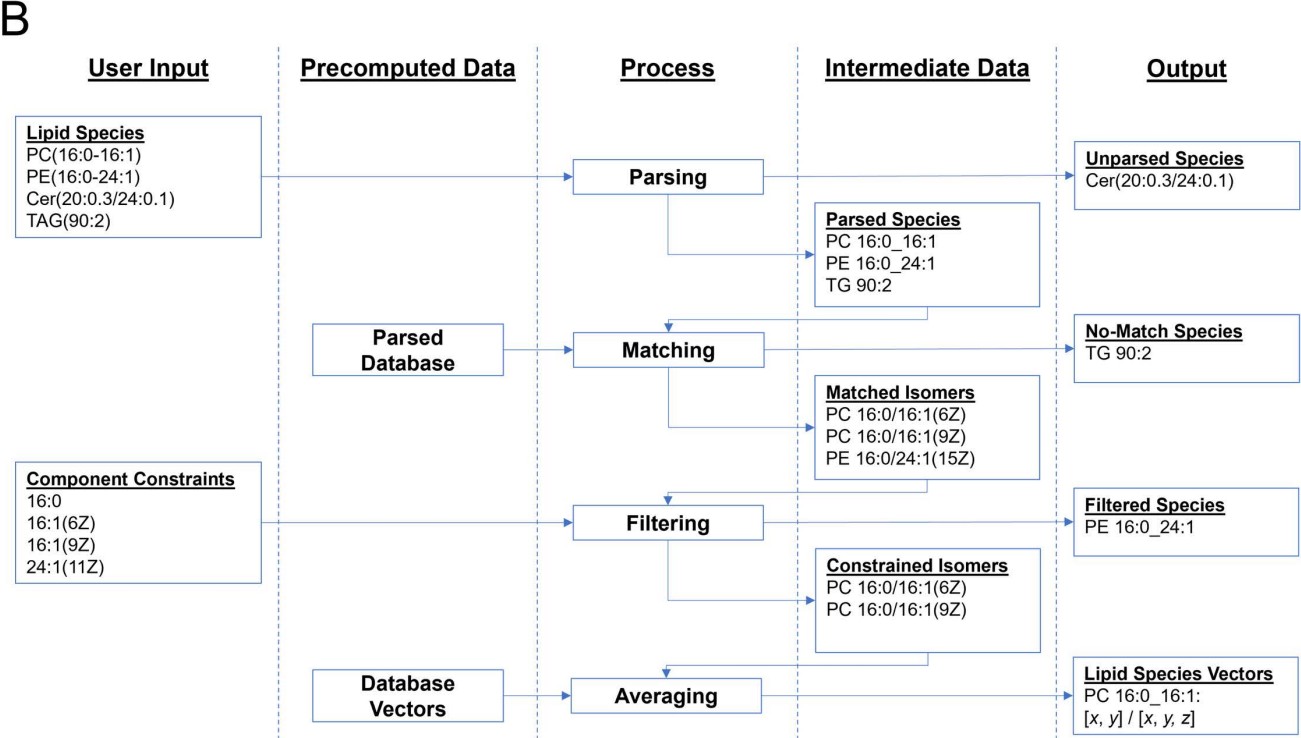

**Fig 1. Vector Space Generation and Matching.** (A) A lipid structure is decomposed into its substructures of different sizes represented by Morgan sparse fingerprint integers, which constitute the training data for Word2Vec. A molecule's vector is the sum of its substructure vectors and is projected to

2D or 3D with stochastic neighbour embedding. (B) The user provides a list of lipid species names and component constraints. Lipid names are parsed and matched to appropriate isomer names from the pre-parsed database. The component constraints are applied to filter the matches. Vectors of the remaining isomers are averaged for each lipid. Not illustrated is an additional step, in which database matching is attempted on the original names of unparsed lipid species.

### Datasets

Publicly available lipidome datasets from drosophila [24], yeast [25] and mouse [26] were used for development and analysed as user cases. Python scripts for the extraction of the original data and formatting into formats appropriate for Lipidome Projector, as well as manually constructed respective FA and LCB constraint files are given in S2 Dataset.

## Results

### Lipid vector space

We first consider the projection of lipids into a vector space by looking at the distributions of points for entries from the combined databases with a valid structure and class. Are the vectors consistent with chemical intuition and database classification? Fig 2A shows the entire lipid set in two dimensions (see S1 Fig for 3D version). With some exceptions, lipids within a category are grouped together in the vector space despite the underlying structural diversity. For the largest categories, glycerolipids (GL), glycerophospholipids (GP) and sphingolipids (SP) a clear separation can be observed with some overlap and outliers at some borders. To look in more detail, one can focus on the class level with the example of selected glycerophospholipid classes. Fig 2B marks three clusters, which largely correspond to diacyl, mono-alkyl and plasmalogen glycerophospholipids respectively. This suggests that the embedding has mostly captured the chemical connectivity at the glycerol. Within each large cluster, phosphatidylinositols (PI) and phosphatidylcholines (PC) form their own subgroups with some local exceptions. For the other classes there are numerous smaller, intertwined clusters spread across the vector space. Also marked are a few unusual molecules with uncommon fatty acyl double bond structures such as (5E, 9E) or chains which are heavily methylated or even contain ladderane, a structural moiety seen in bacteria. These are positioned outside the main group as one might expect since the database is dominated by the biochemistry of mammals. The remaining plots in Fig 2 show how the lipid vectors capture chemical functional groups and their structural context. In Fig 2C there is a general trend of more double bonds from left to right. Focusing on a local region shows that clustering is determined by lipid class (Fig 2D) and fatty acyl double bond location and number (Fig 2E). Additionally, one can see a systematic change in mass as one moves along clusters (Fig 2F). These patterns suggest that the embedding captures gradual structural changes. This was further assessed using a contrived example borrowed from the literature [5]. Three sets of manually generated structures were added to the training data. The first two consist of series of phosphatidylinositols with a successively longer fatty acyl chain. The sets are the same, except for the presence/ absence of a double bond in the lengthening chain. Fig 3A shows that growing an aliphatic chain gives progressively changing vector positions, while the presence of the double bond leads to a large, but consistent displacement. The third set consists of a series of ceramides, each of which is hydroxylated at a different position within its fatty acyl chain (Fig 3B). The steps of the hydroxylated position translate into an almost linear series of vectors with the exception of an outlier near the acyl bond.

Another aspect of the quality of the vector space is its coverage of lipid classes, fatty acyls, and long-chain bases, which in our case, is completely dependent on the underlying databases and the parser. When lipidomes are imported, entries are discarded if they cannot be matched

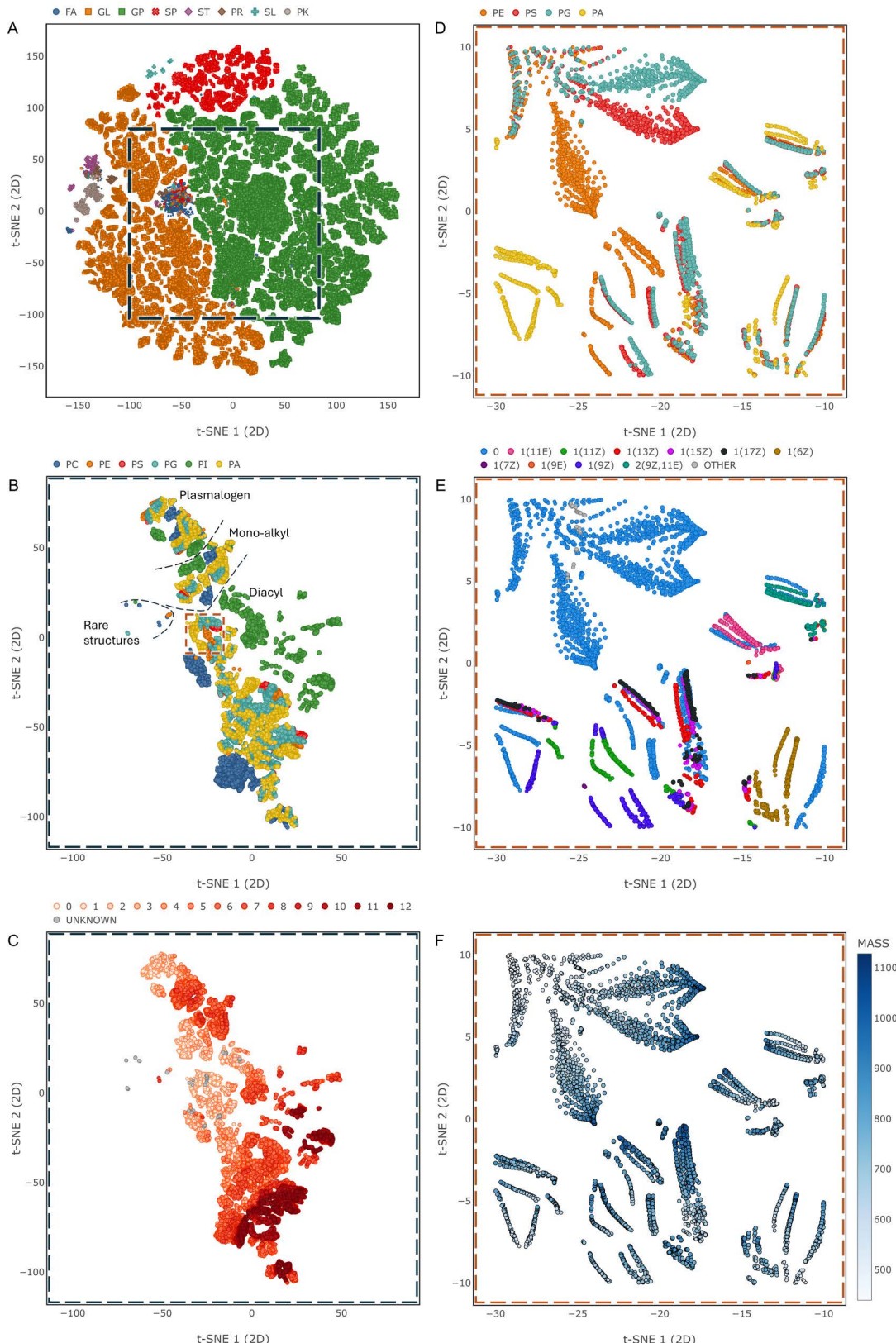

**Fig 2. Vector Space (2D).** (A) Entire vector space. Marker colour represents lipid category: Fatty acids (FA), glycerolipids (GL), glycerophospholipids (GP), sphingolipids (SP), sterol lipids (ST), prenol lipids (PR), saccharolipids (SL) and polyketides

(PK). (B) Region of the vector space focused on selected glycerophospholipids: Glycerophosphates (PA), glycerophosphocholines (PC), glycerophosphoethanolamines (PE), glycerophosphoglycerols (PG), glycerophosphoinositols (PI) and glycerophosphoserines (PS). Marker colour: Lipid class. (C) Same region as in B, marker colour represents the number of fatty acyl double bonds. (D) Zoomed-in region of selected glycerophospholipids, marker colour represents lipid class. (E) Same region as in D, marker colour represents the double bond profile of the 2-sn fatty acyl. (F) Same region as in D, marker colour represents molecule mass. See S3 Dataset for interactive HTML.

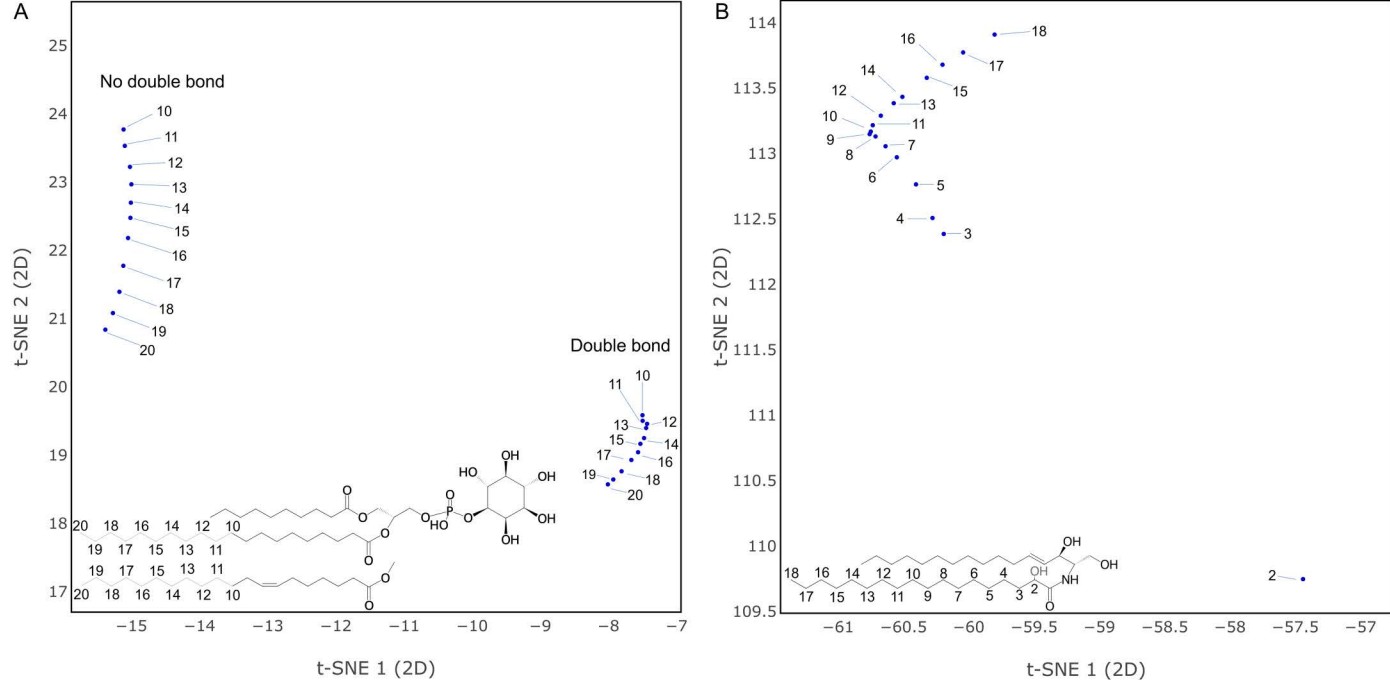

**Fig 3. Impact of Stepwise Structural Changes.** (A) Local vector space region of manually added phosphatidylinositol structures. Marker annotations denote the length of the 2-sn fatty acyl. (B) Local vector space region of manually added ceramide structures. Marker annotations denote the fatty acyl hydroxylation position.

or if they are rejected by the constraint-based filtering. For the three example literature datasets used here, we implemented plausible FA/ LCB constraints and performed the matching to the database. Reasonable manual preprocessing steps, such as re-formatting the data, removing duplicate entries, and adjusting unusual nomenclature were performed beforehand, and are available as Python scripts in S2 Dataset. The processing statistics are listed in Table 1.

## Visualization

One has to look at complete databases to judge the vector space and embedding of lipids. A user, however, would be interested in what can be seen in their lipidome. We take three examples from the literature and look at the scatterplots in the light of the biochemistry noted by the original authors.

The first dataset consists of lipidomes of different *Drosophila melanogaster* larval tissue types (brain, fat body, gut, lipoprotein, salivary gland, wing disc) fed with different diets (plant food or yeast food) [24]. For our quick analysis, we averaged the lipidome samples by tissue type. Carvalho et al noted that hexosyl ceramides (HexCer) and ether glycerophospholipids (O-) were only detected in gut and brain tissues respectively [24]. Fig 4A shows how this kind of feature can be easily observed and highlighted. Fig 4B displays a comparison of fat body and lipoprotein tissue types focused on a glycerolipid region and highlights the expected large

**Table 1. Matching statistics for development datasets.**

| Dataset | Num. lipids | Successfully matched | Parsed - not matched | Not parsed - not matched | Filtered |
|---|---|---|---|---|---|
| *Drosophila* | 359 | 324 (90.3%) | 9 (2.5%) | 4 (1.1%) | 22 (6.1%) |
| Yeast | 249 | 235 (94.4%) | 14 (5.6%) | 0 | 0 |
| LAMP3 | 209 | 199 (95.2%) | 3 (1.4%) | 0 | 7 (3.3%) |

amounts of triacylglycerol (TG) species in the fat body and conversely an overabundance of diacylglycerols (DG) in the lipoprotein tissue, both noted in the original publication.

The second example is focussed on a yeast study comparing the wildtype strain (BY4741) and mutants that were defective in fatty acyl elongation (Elo1, Elo2, Elo3) [25]. Two different growth temperatures (24°C and 37°C) were considered. The study showed that the Elo2 and Elo3 strains produce sphingolipids with shorter fatty acyl chains. We averaged the samples by strain, filtered Elo1, and projected the full results onto our vector space (Fig 4C). Fig 4D displays sphingolipid abundances from the wildtype strain compared to average abundances from the Elo2 and Elo3 group, clearly showing that species with short fatty acyls (=< 22 chain length) occur exclusively in higher amounts in these mutant strains.

The third dataset is taken from a study of LAMP3-deficient mice, evaluating the role of this protein in the lung [26]. The two different conditions genotype (wildtype/ LAMP3-KO) and challenge (none/ allergen induced asthma) resulted in four groups of mice. Fig 4E and 4F show that if we group the samples by genotype and challenge, average the lipids abundance values across samples in each group, and compare the wildtype to the LAMP3-KO genotypes in the asthma group, there is a large reduction in phosphatidylglycerols in the LAMP3-KO group, as noted by the authors. Fig 4E also shows the increased abundance of diacylglycerols and decreased amounts of certain sphingolipids and phosphatidylinositols in the wildtype group.

## Discussion

There are two aspects to this work. Firstly, there is the fundamental embedding of molecules in a low-dimensional space. Secondly, there are practical issues and the software implementation.

From the point of view of the vector space, there are some surprising observations. The lipid coordinates agree with chemical intuition, although the training was completely unsupervised. Not only were classic lipid categories separated, but unusual structures were given coordinates on the edges of the common lipid classes (Fig 2B). The local and global structure of the embedding is interesting. Globally, the space separates broad classes, but locally it reflects chemical detail. It is remarkable that moving a hydroxylation along a chain gives a set of points near each other that appear to lie on a smooth curve. There is reason to say this is unexpected. Consider the space as first calculated in 100 dimensions. Maybe there are directions corresponding to phosphorylation, chain extension, moving bonds and other chemical properties. When we project the space to two or three dimensions, one will inevitably lose information. The local structure is a tribute to stochastic nearest neighbour-embedding rather than any invention on our part.

We must concede that this exercise has little geometrical rigour. The embedding might maintain local relationships in the two-dimensional space, but longer-range distances are compressed or extended. Given the method's emphasis on a point's neighbourhood, a densely populated region in 100-dimensional space is treated differently to a sparse region. Neighbouring regions are likely to end up with inconsistent orientations. One could see this as a weakness [27] but Lipidome Projector is a visualisation tool. One can regard the projections

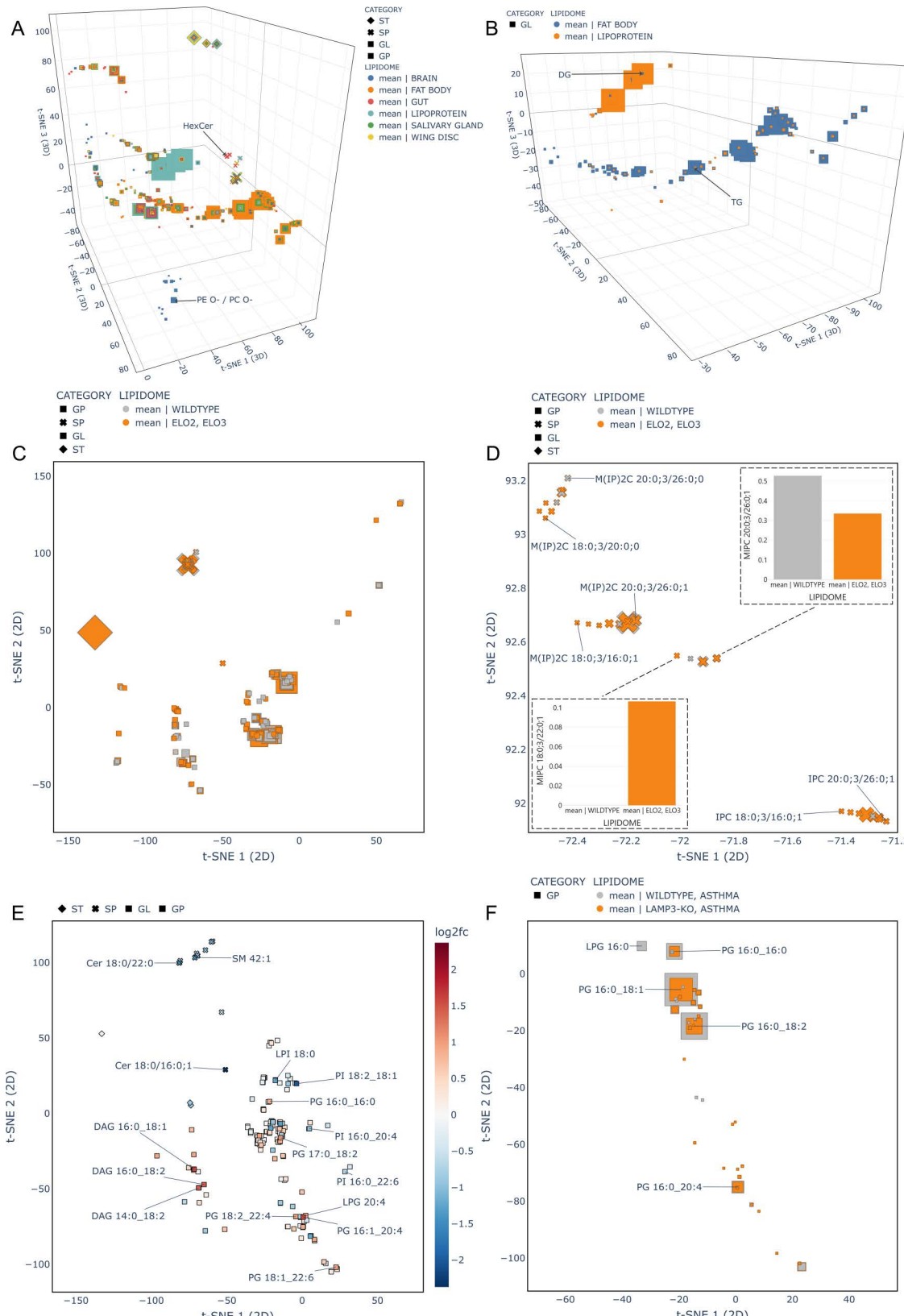

**Fig 4. Lipidome Dataset Projections.** (A) *Drosophila* dataset averaged over tissue type. HexCer and ether-linked GPs are only present in gut and brain tissues respectively. Min-max scaling of abundances was used to calculate marker area. (B) *Drosophila*

dataset zoomed in to a glycerolipid region of the vector space showing selected tissue samples (same marker scaling as in A). (C) Yeast lipidomes – comparison between the means of the wildtype and the Elo2 and Elo3 strains with min-max marker scaling. (D) Yeast dataset zoomed in on a region of partially annotated sphingolipids (same marker scaling as in C). Elo2 and Elo3 strains contain species with shorter fatty acyls. (E) Mouse lung lipidome dataset lipids coloured by the $\log_2$ abundance fold change between the wildtype and LAMP3-KO asthma conditions. Certain lipids with relatively high change values are annotated. (F) PG region comparison between wildtype and LAMP3-KO asthma conditions. Linear scaling applied to marker sizes. See S3 Dataset for inter- active HTML.

as no more than an artistic or practical representation of the higher-dimensional space [28]. Calculations such as cluster analysis or lipidome homology should be done in the 100- dimensional space since this is geometrically our best construction. The embedding also reflects biases in the training set due to the selection of classes present in the chosen databases and their respective sizes. Lipids with more fatty acyls (e.g., Triacylglycerols or Cardiolipins) inherently dominate combinatorially generated datasets. Finally, we know that different molecules always have different coordinates, but since marker sizes are scaled relative to abun- dances, it is inevitable that points will occasionally obscure each other.

There are also differences compared to other vector spaces for lipids. Marella et al calcu- lated the differences between molecules using the distances between string representations of the molecules [5]. This suffers from the instability of string representations. Kopczynski et al avoided this problem by using graph-based similarity [7]. There is a less obvious difference between the methods. Kopczynski et al calculated distances between lipids and used principal component analysis (PCA) to get low dimensional coordinates from the distance matrix [7]. This is deterministic, but discarding everything after the first few eigenvectors is a brutal trun- cation. Applying PCA to our 100-dimensional coordinates, we cover only 56% of the variance with the first two principal components and only 71% with the first three. 90% of the variance is only reached with the first 15 principal components. Clearly, the intrinsic dimensionality of the 100-dimensional space is higher than the two or three dimensions we reduce it to. To make this point, one can estimate the intrinsic dimensionality with an implementation [29] of an established PCA-based approach [30]. Here, the intrinsic dimensionality is the number of normalized eigenvalues larger than a threshold value (here 0.05). This yields 6. The same computation performed for each individual vector and its 100 nearest neighbours results in an average intrinsic dimensionality of 12.5. Projecting the previously considered region of glycerophospholipids onto the first principal components confirms the noted general trends of organization by number of fatty acyl double bonds (S2C Fig) and mass (S2D Fig). Lipid classes, however, overlap entirely in two dimensions (S2A Fig) and only begin to separate into distinct slices with the inclusion of the third (S2B Fig). One can say that t-SNE is a compro- mise, but in the light of these results, it is effective in conveying different influences (class, fatty acyl features, mass) in a low dimensional representation.

Kopczynski et al's approach does admit one feature that we lack. We construct a space based on all known lipids and then show all lipidomes in this context. In contrast, Kopczysnki et al build a new space for each set of lipidomes [7]. This allows them to construct a very natu- ral measure for the similarity of lipidomes and lends itself to clustering of datasets.

Continuing in this self-critical vein, the non-determinism of our approach might be con- sidered a disadvantage. Repeating the training and dimensional reduction always gives slightly different results. With more training time or different parameters, one might get even better results. Having experimented in this direction, we suspect that this is not a useful pursuit. It would be more profitable to consider completely different strategies. We see graph convolu- tional networks as a more natural fit to molecular structures [31] and one could experiment with novel dimensionality reduction methods such as UMAP [32].

Besides the embedding, other issues should be addressed. We are not the first group to lament the inherent inconsistency of lipid nomenclature [19]. Synonyms such as SM(d18:1/14:0) and SM 18:1;2/14:0 are tedious but can be handled mechanically by packages such as Goslin. A more fundamental problem are lipid notation ambiguities which cannot be solved by any parser.

In this study we encountered ambiguities in the position, number and precise location of double bonds and hydroxylations of sphingolipids. Some line notations would allow one to denote some ambiguities [33], but lipidome data is typically not stored in such formats. Another problem is that a user lipidome may contain species that are not in the training set (SwissLipids + LMSD). This problem will be alleviated when we implement an on-the-fly method to generate structures and respective vectors from nomenclature only.

The second half of this work is the software. With the vector space precomputed, it is not too demanding to run on an ordinary laptop. The web application stores lipidome data on the client side and sends it to the server for processing operations. This does require a fair amount of client-server communication, but we are currently moving more processing tasks to the client's browser. User interfaces and encoding data are also a matter of taste. For example, we concede that the compact representation of relative abundances might seem foreign to a user.

There are clear directions for the future. There will be improvements to the underlying vector space as we experiment with the embedding model and as the databases are updated. The software also benefits automatically from the evolution of the Goslin parsing package [20]. The interface and display straddle taste and usability. A colour-blind-friendly palette is necessary, as is overlap removal. Different kinds of abundance displays will improve with more user feedback. Finally, we plan proper integration with biochemical pathway software. As it stands, the vector space is conceptually useful, and the software fills a practical niche.

## Supporting information

**S1 Fig. Vector Space (3D).** (A) Projection of the entire vector space. Marker colour represents lipid category: Fatty acids (FA), glycerolipids (GL), glycerophospholipids (GP), sphingolipids (SP), sterol lipids (ST), prenol lipids (PR), saccharolipids (SL) and polyketides (PK). (B) Region of the vector space focused on a set of selected glycerophospholipids: Glycerophosphates (PA), glycerophosphocholines (PC), glycerophosphoethanolamines (PE), glycerophosphoglycerols (PG), glycerophosphoinositols (PI) and glycerophosphoserines (PS). Marker colour: Lipid class. (C) Same region as in B. Marker colour: Number of fatty acyl double bonds. (D) Zoomed in region of selected glycerophospholipids. Marker colour: Lipid class. (E) Same region as in D. Marker colour: Double bond profile of the 2-sn fatty acyl. (F) Same region as in D. Marker colour: Molecule mass. See S3 Dataset for interactive HTML. (TIF)

**S2 Fig. PCA Vector Space Region.** (A) Region of the vector space focused on a set of selected glycerophospholipids: Glycerophosphates (PA), glycerophosphocholines (PC), glycerophosphoethanolamines (PE), glycerophosphoglycerols (PG), glycerophosphoinositols (PI) and glycerophosphoserines (PS). Axes correspond to the first two principal components. (B) Same region and colours as in A. Axes correspond to the first three principal components. (C) Same region and axes as in B. Marker colour: Number of fatty acyl double bonds. (D) Same region and axes as in B. Marker colour: Molecule mass. See S3 Dataset for interactive HTML. (TIF)

**S1 Table. Word2Vec Embedding Parameters.** (DOCX)

**S2 Table. Stochastic Neighbour Embedding Parameters.**
(DOCX)

**S1 Dataset. List of classes present in LMSD and SwissLipids recognised by the Goslin parser in translated representation.**
(ZIP)

**S2 Dataset. Python scripts with instructions for the extraction and transformation of original datasets; Transformed datasets; Dataset FA/ LCB constraints.**
(ZIP)

**S3 Dataset. Partially interactive HTMLs of vector space and dataset projection scatter plots.**
(ZIP)

## Acknowledgments

We are indebted to Dr Nils Hoffmann (Forschungszentrum Jülich GmbH, IBG-5, Jülich, Germany) for advice and consultation beyond the call of reasonable duty. Dr Dominik Kopczynski (University of Vienna, Department of Analytical Chemistry, Vienna, Austria) provided invaluable insights on many technical issues.

## Author contributions

**Conceptualization:** Timur Olzhabaev, Dominik Schwudke, Andrew Ernest Torda.

**Data curation:** Daniel Krause, Dominik Schwudke.

**Funding acquisition:** Dominik Schwudke.

**Investigation:** Timur Olzhabaev, Lukas Müller.

**Methodology:** Timur Olzhabaev, Lukas Müller, Daniel Krause.

**Resources:** Andrew Ernest Torda.

**Software:** Timur Olzhabaev, Lukas Müller.

**Supervision:** Dominik Schwudke, Andrew Ernest Torda.

**Validation:** Timur Olzhabaev, Lukas Müller.

**Visualization:** Timur Olzhabaev, Lukas Müller, Daniel Krause.

**Writing – original draft:** Timur Olzhabaev, Andrew Ernest Torda.

**Writing – review & editing:** Lukas Müller, Dominik Schwudke, Andrew Ernest Torda.

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
