## [Decision Letter · Decision Letter 0]

4 Oct 2024

Dear Mr. Olzhabaev,

Thank you very much for submitting your manuscript "Lipidome visualisation, comparison, and analysis in a vector space" for consideration at PLOS Computational Biology.

As with all papers reviewed by the journal, your manuscript was reviewed by members of the editorial board and by several independent reviewers. In light of the reviews (below this email), we would like to invite the resubmission of a significantly-revised version that takes into account the reviewers' comments.

We cannot make any decision about publication until we have seen the revised manuscript and your response to the reviewers' comments. Your revised manuscript is also likely to be sent to reviewers for further evaluation.

Sincerely,

Iddo Friedberg, Ph.D.

Academic Editor

PLOS Computational Biology

Pedro Mendes

Section Editor

PLOS Computational Biology

Reviewer's Responses to Questions

**Comments to the Authors:**

Reviewer #1: This is an excellent paper that reports on a very interesting study. The authors build on machine learning methods developed for text analysis to obtain an embedding of lipids. They show that the embedding is meaningful, suggesting that all data points corresponding to different lipid species lie on a data manifold. The embedding is pre-computed, so that specific lipidomic data sets can be easily projected in the embedding to obtain a low-dimensional and insightful representation of the data set. The authors use their method to re-analyze lipidomic datasets, and show that the embedding is useful to recover biological insight. The authors provide an open source software to the community.

I like this paper very much, it's well written, the procedure is well-described, and the results are convincing. I would however be grateful if the authors could consider the two following comments:

1. t-SNE is notoriously unstable and can provide structure in data even when there isn't. The structure identified by the authors in, for instance, Fig. 2A has a clear semantic interpretation, which suggests the clusters are actually in the data. however, it would be very good if the authors could also use an orthogonal dimensionality reduction technique, eg diffusion maps or UMAPS, to verify if they get clusters and if these clusters are compatible with those emerging with a t-SNE based visualization.

2. The embedding obtained by the authors is very intriguing, as it suggests that the lipid fingerprints lie on a low-dimensional manifold, and that the embedding extracts general features that describe the intrinsic physiochemistry of the lipids. The authors could try to use several methods to estimate the intrinsic dimensionality, say perform PCA locally and estimate how many components are necessary to retain 90% of the variance, and then take an average. Other methods exist.

Reviewer #2: The authors describe a spatial embedding methodology and its implementation as a new interactive browser based visual analytics tool designed for exploring datasets from lipidomics experiments. The tool is distributed as a python based server that can also be easily run as a standalone system on a private computer, and employs standard web technologies that should enable it to be provided as an online resource.

The spatial embedding is constructed using an adapted mol2vec neural network - the idea being that chemical similarity should be preserved in the 2D/3D embeddings, allowing encoding of both lipid class and precise molecular signatures that result from lipidomics (e.g. chemical modifications). 2D and 3D embeddings are created with tSNE, and sets of observed lipidome data mapped to their locations by matching names (or in the case of MS data, lipid identification + constraint set) to the corpus of names associated with mol2vec vectors. Example datasets are provided based on published work. The authors first demonstrate how the embedding captures the semantics of lipid chemical space through a series of 'spike in' synthetic structures, then show how their tool allows visualisations that capture the same insights as identified by authors of existing lipomics studies.

Questions/major points of revision

1. The novel approach for creating a low dimensional embedding of lipid space lies at the core of this work. However, embeddings created with tSNE can be misleading. Have the authors considered whether the 'happy accidents' of tSNE (lines 287-292) are realistic depictions of the 100-dimensional space ? See for example https://ieeexplore.ieee.org/document/9064929 (rxiv version is at https://arxiv.org/pdf/2002.06910). The authors should also consider framing their discussion in the light of Chari & Pachter's strong rebuttal regarding the use of embeddings for interpreting single cell genomics (https://journals.plos.org/ploscompbiol/article?id=10.1371/journal.pcbi.1011288). Whilst the approach here is quite distinct, much of the computational biology community are now aware of the limitations of these approaches, and clear articulation of the reasons why tsne is a good choice here is important.

2. The authors note in line 104 that mol2vec's approach was modified to distinguish enantiomers. Does Mol2Vec truly ignore chirality ? I was suprised at how little detail was provided regarding how mol2vec's approach was modified - it should at least warrant a reference.

3. In line 184, the authors note the embedding organises lipids with a 'general trend of more double bonds from left to right' - this could be quantitated (by pca probably), or at least visualised by summing the number of double bounds in each molecule in a sliding window across the embedding's space from left to right. Similarly, line 187's statement regarding mass distributions could be more rigourously treated if it is considered germane.

4. A more pressing question is - are these apparent organisations of the embedding really useful for visualisation ? e.g. whilst trends in mass seem to be visualised within a class, there is no consistent orientation across all classes. It is arguable that people will gain familiarity with the landscape of the embedding (in the same way as one might learn the layout of a complex pathway map, or genome), but is there, in that case, another embedding that might even more effectively itself to that ?

5. The training data spikes do effectively demonstrate that the embedding is sensitive to differences in chemistry. What would be also interesting to measure is how sensitive the rest of the lipidome embedding is to additions - are other regions unaffected by the addition of these new species ?

6. I note a number of points of discussion regarding the Carvalho dataset:

i. Fig 4A & B show clear delineation, but is there similarly clear separation with the 2D embedding ? 3D is often impractical for publication (and generally frowned upon by dataviz experts) so it would be informative to the general reader to see the 2D equivalents - either as a figure in the main manuscript or referred to in supplementary.

ii. Is it possible to assign different glyphs to DG and TG rather than needing to label them ?

iii. Can the tool support display of the differences in lipidome between conditions in the experiments performed by Carvalho ?

7. Embeddings limit visualisation of differences in abundance across a set of observed entities (ie lipids in this case): Figure 4C in particular demonstrates this limitation. The inset histograms in the zoomed in region (4D) are vastly superior, but this approach also takes up lots of space on the page. Have the authors considered exploring the use of small multiples to show the differences, or visualisation of signed differences using a colourscheme, overlaid on grey glyphs sized according to wild type abundances ?

8. Differential abundance analysis & Statistical rigor. It is unfortunate that the original publication does not provide the results of their own statistical analysis in a machine readable form (the lipidomeX link no longer works). This tool could thus make a very useful contribution if it were to offer such a calculation in an easy to use (and reproducible) manner. The visualisation in Figure 4E is a good start, but this visualisation is very difficult to understand without some indication of which changes are actually significant.

i. Line 268-269: "show that if we average the samples by genotype and challenge" - here you presumably mean 'average the samples for the same genotype in each condition' ? It's important to be clear, because the original paper demonstrates variance is negligible across replicates, so it is sufficient to compare the two genotype averages for the challenge condition.

ii. It is not clear from the manuscript that the foldchange plot is actually computed with lipid explorer. You should really make this clear! I managed to recreate the figure myself, but it might be useful to include some simple instructions on how to recreate the figure (also see point 7).

iii. Can statistical support be calculated for foldchange ? Ideally it should take into account experiment design (e.g. technical & bio-replicates) - but either way, it is important to be able to identify significance thresholds, or at the very least, provide a volcano plot.

iv. If statistical support can not be included, is there perhaps a way to import the results of an analysis performed elsewhere (here, data import/api documentation will help!)

9. Reproducibility. The authors provide all the scripts necessary to create/recreate example data sets as supplementary, and also stand-alone HTML documents produced by the tool for each figure in the paper. However, whilst the export options that the tool provides are effective, and certainly necessary, I could not see a way to save the session so it could be restored - this capability is essential for a tool that will be used to create figures for papers.

i. This limitation should be explicitly noted if it cannot be addressed in reasonable time.

10. Acknowledge limitations of the visualisation capabilities of the tool. The tool appears to only allows quantitative data (abundances, fold-change, etc) to be encoded as colour, or glyph size when visualising sets of lipidomes. Many of the figures demonstrate that occlusion occurs when lipidomes are shown as circular glyphs, and additionally encoding quantities as glyph size only makes the occlusion problem worse.

i. Ideally, visualisations of these data should be properly evaluated by a user study to ascertain whether it is actually effective for people from a variety of backgrounds. I am certain that the scaled glyph approach is far from ideal, but the nature of the embedding may lend itself to other approaches. Complex or difficult to perceive colourschemes should also be avoided. At the very least, the authors could include a few sentences describing how visualisations might be improved in the future.

ii. If possible, explore how the differences shown in 4F could be represented in a more effective way.A better approach might be to use the wild-type as a 'background' - shown in grey, and then simply highlight the difference with a single colour, rather than using two sets of coloured glyphs. Alternatively, employ a glyph that can encode two different quantities without occupying varying amounts of space: piecharts are often used for this, but other shape based approaches exist.

iii. If P-values were available, then complexity could also be reduced by simply applying a threshold to exclude insignificant differences.

Minor

M1. The authors need to be more clear in their abstract and introduction that they evaluate the effectiveness of their approach using existing published datasets, and examine whether their approach allows recapitulation of the conclusions reached by the authors.

M2. line 111. The authors note each fragment derived from the additional descriptors employed in the mol2vec approach were used to create 'separate training sentence'. Has this approach employed by others and validated rigorously ?

M3. line 182 - you should mention that figures 2C-2F are the zoomed in region of figure 2A (presumably indicated by the orange border in those plots). I would also suggest avoiding using a colour for the border that is also employed in a scale (e.g. in 1F).

-- Figure 1F's mass scale should employ just a single colour, where saturation encodes mass - since the one chosen is more suited to signed quantitites such as temperature.

M4. line 193/196 - Better to swap panels for Figure 3 so the chain length/double bond spike set are labelled 3A, if they are best described first rather than second.

M5. line 230: "A user, however, would be interested in what one sees in their lipidome" - perhaps better to say '..what can be seen in their..'

M6. line 263: 'occurring' - should be 'occur'

Notes and suggested improvements to the lipid projector tool

F1. Documentation.

i. the github repository's readme provides a very brief summary of how to get going, and cites external data employed in the development of the tool. It also cites papers with example datasets - stating that they were used in the paper. Here the authors should also make it clear that these example datasets are also included in the github repository.

ii. the authors include in the lipidome_projector/scripts/vector_space/README.md details of how the embedding was constructured. However, they should also either include the data used to create the accommpanying embedding, or at least a link and checksum for the data that was used and where it was retrieved. Presumably it was https://www.lipidmaps.org/files/?file=LMSD&ext=sdf.zip ?

iii. The 'load data' tour was very effective as an onboarding experience. However, I would also suggest providing some brief text based documentation in the git repository regarding the expected format of data files that can be imported - for instance, one might wish to start the tool with a set of data located on the same machine, but more likely, someone is preparing data via a notebook environment.

F2. In the Graph settings panel: does linear vs min-max scaling have any real use ? They appear to change the size of the glyphs, but visually there's not much impact (this relates to my point above regarding visual encodings: size of glyph is far less effective than position).

F3. Suggest default download be SVG (best for publication) - or provide a couple of options in the popup for a diagram.

F4. Consider ways that visual complexity can be reduced:

* Use of 3D is clearly necessary when viewing both differential abundance & diversity changes, particularly since t-sne 2D plots are obviously congested, however, there is a serious visual overload issue with the default encodings. Plotly.js is a very powerful multidimensional visualisation platform, and the authors have built a very sophisticated package on top of it, but the fundamental barrier remains the complexity of lipidome data - which with this tool, often yields rather interesting cubists figures instead of intuitively understandable diagrams.

* The complexity of the form/card style user interface for creating groupings, assigning colours, doesn't really lend itself to navigating the data, since selection is primarily via the grid - which works if someone knows what they are looking for but not otherwise: It would be great if the Key allowed interactive selection of groups/subsets - allowing them to be navigated to in the table, and enable/disable their display in the plot, which would really help simplify visualisations.

F5. From a chemists perspective, representative structure is really important. Would it be possible to add sticky labels with the representative structure for a progression (as is kind of shown in the paper ?). It would also be useful to be able to visualise a fleet of structures in the selected region, annotated with abundances/differences across conditions.

F6. For comparisons, small multiples might be useful to complement the quantitative visualisations of compositional statistical changes.

F7. As already noted earlier - the app itself appears to have no 'Save Session' function - this is essential for any data analysis tool, particularly one that also enables figures or other visualisations to be generated - since it is critical to be able to return to the data used to generate the figure.

Reviewer #3: The review was uploaded as attachment.

**Have the authors made all data and (if applicable) computational code underlying the findings in their manuscript fully available?**

Reviewer #1: Yes

Reviewer #2: Yes

Reviewer #3: Yes

PLOS authors have the option to publish the peer review history of their article (what does this mean? ). If published, this will include your full peer review and any attached files.

**Do you want your identity to be public for this peer review?** For information about this choice, including consent withdrawal, please see our Privacy Policy .

Reviewer #1: No

Reviewer #2: No

Reviewer #3: No
---

## [Decision Letter · Decision Letter 1]

9 Jan 2025

PCOMPBIOL-D-24-01371R1

Lipidome visualisation, comparison, and analysis in a vector space

PLOS Computational Biology

Dear Dr. Olzhabaev,

Thank you for submitting your manuscript to PLOS Computational Biology. After careful consideration, we feel that it has merit and will be acceptable provided a minor revision. Therefore, we invite you to submit a revised version of the manuscript that addresses the one point raised during the review process.

Please submit your revised manuscript within 30 days Mar 11 2025 11:59PM. If you will need more time than this to complete your revisions, please reply to this message or contact the journal office at ploscompbiol@plos.org. Please include the following items when submitting your revised manuscript:

We look forward to receiving your revised manuscript.

Kind regards,

Pedro Mendes, PhD

Section Editor

PLOS Computational Biology

**Additional Editor Comments:**

Please see if you can accept the suggestion to reword a sentence. Nothing else stands out right now, and we will be ready to accept the manuscript once you decide on that sentence.

**Journal Requirements:**

1) Some material included in your submission may be copyrighted. According to PLOSu2019s copyright policy, authors who use figures or other material (e.g., graphics, clipart, maps) from another author or copyright holder must demonstrate or obtain permission to publish this material under the Creative Commons Attribution 4.0 International (CC BY 4.0) License used by PLOS journals. Please closely review the details of PLOSu2019s copyright requirements here: PLOS Licenses and Copyright. If you need to request permissions from a copyright holder, you may use PLOS's Copyright Content Permission form.

Potential Copyright Issues:

i) The following Figure contains screenshots: Figure S1.. We are not permitted to publish these under our CC-BY 4.0 license, websites are usually intellectual property and are copyrighted.This includes peripheral graphics of the web browser such as icons and button. We ask that you please remove or replace it.

**Reviewers' comments:**

Reviewer's Responses to Questions

**Comments to the Authors:**

Reviewer #2: I thank the authors for consideration of my review requests, and note the robustness of their response accompanying their revised manuscript and software.

The additional discussion and PCA-based analysis adds weight to their validation of the properties captured by the 100-dimension space, and most importantly the note recommending users do not employ the 2D embedding's coordinates directly for clustering/etc. The software's documentation is also much improved, and I look forward to seeing how it will be used and developed in future - particularly in the integration with pathway visualisations and other software.

I have one minor revision, however. The authors' observation that 'software is a matter of taste' is perhaps overly subtle. They should instead be more direct. I suggest the following rewording:

l 367-370. "Software is also a matter of taste. The current release displays properties such as relative abundances using very compact methods, but these might at first seem foreign to a user."

Could instead be:

"We have also chosen to display properties such as relative abundances using very compact methods. User interface design and choice of visual encodings are often a matter of taste, and the ones we chose may seem less optimal to some users."

**Have the authors made all data and (if applicable) computational code underlying the findings in their manuscript fully available?**

Reviewer #2: Yes

PLOS authors have the option to publish the peer review history of their article (what does this mean? ). If published, this will include your full peer review and any attached files.

**Do you want your identity to be public for this peer review?** For information about this choice, including consent withdrawal, please see our Privacy Policy .

Reviewer #2: **Yes: ** Jim Procter

**Figure resubmission:**
---

## [Editor Report · Decision Letter 2]

20 Feb 2025

Dear Mr. Olzhabaev,

We are pleased to inform you that your manuscript 'Lipidome visualisation, comparison, and analysis in a vector space' has been provisionally accepted for publication in PLOS Computational Biology.

Best regards,

Pedro Mendes, PhD

Section Editor

PLOS Computational Biology

---

## [Editor Report · Acceptance letter]

PCOMPBIOL-D-24-01371R2

Lipidome visualisation, comparison, and analysis in a vector space

Dear Dr Olzhabaev,

I am pleased to inform you that your manuscript has been formally accepted for publication in PLOS Computational Biology. Your manuscript is now with our production department and you will be notified of the publication date in due course.

With kind regards,

Anita Estes
